# Evaluating Single-Cell Foundation Models for Cell Retrieval

## Abstract

Efficiently and accurately searching large-scale single-cell RNA-seq databases has been a long standing computational challenge. There is an increasing number of single-cell retrieval methods, particularly those based on single-cell foundation models, proposed in the literature. However, this field lacks a comprehensive benchmark among these methods. This gap exists due to the lack of standard evaluation metrics and comprehensive benchmark datasets. Addressing these challenges, we propose a comprehensive evaluation benchmark to assess the capabilities of 12 existing single-cell retrieval methods from three classes: non-machine learning method, VAE-based methods and single-cell foundation model (scFM) based methods. We propose a series of label-dependent and label-free evaluation metrics to assess the performance of single-cell retrieval methods. Through benchmarking across diverse settings (cross-platform, cross-species and cross-omics), our notable findings include: top scFMs such as UCE, scFoundation and SCimilarity show substantial overall advantage compared with other methods; traditional non-machine learning method perform well in cell retrieval thus should not be neglected; common cells retrieved by top methods share distinct gene expression patterns; label-free metrics have consistent evaluation outcome compared with label-based methods thus can be employed in a broader scenario. Our rigorous and comprehensive evaluation identifies the challenges and limitations of current retrieval methods and serves as foundation for further development of single-cell retrieval methods.[1]

## 1 Introduction

The technological advancements in single-cell sequencing has led to increasingly available scRNA-seq datasets. To date, the transcriptome of trillions of cells spanning diverse tissues and species have been profiled using various sequencing technologies (Biology et al., 2023). The main objective of vast scRNA sequencing is to create comprehensive single-cell reference for discovering new biological insights and serve as foundation for clinical usage. For example, for biologists, they could utilize existing single-cell references to identify cell-type specific marker genes or disease cell states (Anders & Huber, 2010). For clinicians, they could compare the sequencing outcomes from patients with the single-cell reference atlas to determine patient specific transcriptome change thus conduct precision diagnosis (Dann et al., 2023).

Both the clinical and biological applications with respect to single-cell reference require easy and fast access to these datasets. However, searching the large scale single-cell reference spanning multiple tissues, species or sequencing platforms is a challenging problem. First, scRNA-seq datasets are usually high dimensional and sparse (Kharchenko, 2021), thus searching with cosine similarity or L2-distance between count vectors would be inefficient and inaccurate. Second, scRNA-seq datasets are largely affected by batch effects (Haghverdi et al., 2018), which means that datasets from different experiment batches or platforms may have substantial distribution shift. Therefore, retrieving from massive single-cell datasets from different experiment platforms is considerably difficult. Furthermore, scRNA-seq datasets are affected by multiple types of technical noise during sequencing (Mereu et al., 2020), thus the retrieval method must be robust.

---

[1]Codebase and datasets will be available upon acceptance.

Figure 1: Pipeline of benchmarking. Single-cell database spanning multiple tissues, species and experiment platforms is constructed. Given query cells, non-machine learning methods, VAE-based methods and single-cell foundation models are used for retrieval. In the cell embedding stage, the count vectors or learned dense embeddings will be used by different methods. In the cell retrieval stage, local sensitive hashing (LSH) and dense retriever are used to retrieve top similar cells with respect to query cell from single-cell database. We benchmark the performance of three classes of cell retrieval methods using both label-dependent and label-free methods.

Multiple methods have been proposed to resolve this challenging problem. These methods can be categorized into three classes, **non-machine learning methods, VAE (Variational Autoencoder) based methods and single-cell foundation model (scFM) based methods.** Initial attempts (Sato et al., 2019; Lee et al., 2021) utilized classical approaches such as local sensitive hashing to efficiently search large-scale scRNA-seq databases. With the development of advanced dimension reduction approach such as VAE (Kingma, 2013), there are growing interests in learning low dimensional representation from high dimensional sparse scRNA-seq count matrix (Lopez et al., 2018; Svensson et al., 2020). Therefore, several single-cell retrieval methods (Cao et al., 2020) have also been developed to search for similar cells using learned low dimensional embedding. Recently, inspired by the advances of large-scale pre-trained foundation models in a wide range of biological domains (Chen et al., 2022; Rao et al., 2020; Fan et al., 2024c), there are also several single-cell foundation models (scFMs) (Theodoris et al., 2023; Cui et al., 2024) developed. These powerful foundation models generate meaningful dense embeddings in a zero-shot manner and have been applied to a wide range of downstream applications, including searching large databases (Heimberg et al., 2023).

Despite numerous efforts in developing powerful methods for searching across large scale single-cell databases, there lacks unified and comprehensive evaluation and benchmark on the effectiveness of these methods. First, there is **no direct comparison between scFMs and other methods in existing works**. SCimilarity (Heimberg et al., 2023) is the only scFM that explicitly performs cell retrieval in the downstream applications, but the performance against other retrieval methods has not been explicitly evaluated. Meanwhile, **the evaluation metrics of cell retrieval are still limited**. Unlike query-passage retrieval in text-mining which has explicit ground-truth annotation on retrieval pairs, there are no ground-truth annotation on cell pairs. Thus, the cell retrieval methods are usually evaluated on whether the retrieved cells have similar cell types as the ground-truth annotation, which can be quite limited as the cell type annotations only provide coarse-grained information and can be biased or incorrect. Furthermore, existing cell retrieval methods have only been **evaluated in limited species or platforms**, which limits their applicability and generalizability. For example, SCimilarity is solely evaluated on scRNA-seq datasets generated with 10x platform for humans and Cellfishing.jl (Sato et al., 2019) is only evaluated on single-platform datasets.

In this paper, we systematically benchmarked the performance of cell retrieval of various single-cell FMs against traditional non-machine learning cell retrieval methods and VAE based retrieval methods with our proposed metrics (Figure. 1). We included 2 non-machine learning methods, 3 VAE-based methods and 7 scFM-based methods, which to the best of our knowledge covers all the major methods for cell retrieval. Our evaluation datasets span multiple-platforms, multiple-species and multiple-omics for unbiased and fair evaluation of cell retrieval. Meanwhile, We designed a comprehensive evaluation pipeline for cell retrieval, including 1) label-dependent evaluation including cell type matching, batch mixing and recall 2) label-free evaluation including the consistency between retrieved cells and the consistency between the DE genes identified from the retrieved cells across methods.

The key takeaways from the comprehensive benchmarking of cell retrieval methods can be summarized as follows.

- Top scFMs such as SCimilarity, UCE and scFoundation show significant advantage over other baseline methods in a zero-shot manner in the majority of settings, but still perform poorly when used on datasets distant from the pre-training database (e.g. on tissues, species or omics unseen or rare during pre-training).

- Traditional non-machine learning methods perform surprisingly well in most settings, highlighting their unique advantage in retrieving cells with similar cell states thus should not be ignored in future benchmarking studies.

- Common cells retrieved by top single-cell retrieval methods share distinct differential gene expression patterns, showing that top methods can identify similar cell states from the reference cells.

- Label-free metrics are consistent with label-dependent metrics, which can be a complementary evaluation method when the cell annotations are missing or inconsistent across different reference datasets.

## 2 Cell Search and Retrieval Methods

### 2.1 Non-machine learning method

Even before the popularity of machine learning in biological data analysis, there are already a number of methods proposed to resolve the cell search and retrieval challenge. CellFishing.jl (Sato et al., 2019) converts the gene expression count matrix to bit vectors with local sensitive hashing (LSH) to perform retrieval. scFind (Lee et al., 2021) searches large scale scRNA-seq database to identify the set of cells that express a set of genes specified by the user.

### 2.2 Variational Autoencoder based Method

Projecting high dimensional and sparse gene expression count vector to low-dimensional dense cell embedding is a central task in machine learning for single-cell analysis. Learning meaningful cell embeddings serves as foundation for a wide range of downstream tasks, such as cell type annotation (Xu et al., 2021) and trajectory inference (Qiu et al., 2017). Therefore, a number of machine learning methods mainly based on variational autoencoder have been developed to learn low-dimensional cell embedding for cell search and retrieval, including CellBlast (Cao et al., 2020), scmap (Kiselev et al., 2018), scVI (Lopez et al., 2018) and LDVAE (Svensson et al., 2020).

### 2.3 Foundation Model based Method

single-cell Foundation Models (FMs) have much higher model capacity than traditional machine learning methods and are pre-trained on massive scRNA-seq datasets. Therefore, single-cell FMs can generate meaningful low-dimensional embedding of cells, even in a zero-shot manner (Heimberg et al., 2023). There are several single-cell FMs have been proposed, including Geneformer (Theodoris et al., 2023), scGPT (Cui et al., 2024), UCE (Rosen et al., 2023), scFoundation (Hao et al., 2024), scMulan (Bian et al., 2024) and SCimilarity (Heimberg et al., 2023). Among all these methods, SCimilarity is the only FM specifically highlighting cell retrieval, while other scFMs are also capable of retrieving cells and the performance of different scFMs has not been explicitly evaluated and benchmarked.

## 3 Evaluation Methods

### 3.1 Problem Definition

Given a set of query cells $\{x_i\}_{i=1}^{N_q}$, the objective is to retrieve top $K$ similar cells $\{\{y_{ik}\}_{k=1}^{K}\}_{i=1}^{N_q}$ from the reference cells $\{y_i\}_{i=1}^{N_r}$. The cells from query cells and reference cells may have annotations from different aspects, such as cell types and experimental batches. Meanwhile, for some other

datasets with multi-modal parallel profiling (i.e. single-cell multiomics-profiling (Baysoy et al., 2023)), there are direct annotations on paired cells (i.e. whether $x_i$ and $y_i$ are measurements of the same cell). Analogs to *information retrieval*, the main objective of single-cell retrieval is to retrieve similar cells from the reference cells (*passages*) given the query cells (*question*), while the success of retrieval can be evaluated by the agreement between labels (*precision and recall*) or downstream label-free applications (*REALM (Guu et al., 2020) and RAG (Lewis et al., 2020)*).

## 3.2 EVALUATION CRITERIA

### 3.2.1 LABEL-DEPENDENT EVALUATION

**Cell Type Accuracy**    The cell states and identities are mainly defined by their cell types, thus evaluating the abilities of retrieval methods to search for cells belonging to the same cell type is of vital importance. We used two metrics to evaluate the capabilities of single-cell FMs in cell type search accuracy, namely average accuracy (`Avg-Acc`) and voting accuracy (`Vote-Acc`). Assume the cell type labels of the retrieved cells are $\{\{l_{ik}\}_{k=1}^{K}\}_{i=1}^{N_q}$ and the cell type labels of the query cells are $\{l_i\}_{i=1}^{N_q}$. We denote the mode of $\{l_{ik}\}_{k=1}^{K}$ as $\mathrm{mode}(\{l_{ik}\}_{k=1}^{K})$. The vote accuracy is computed by $\texttt{Vote-Acc} = \frac{\sum_{i=1}^{N_q} \mathbb{I}(\mathrm{mode}(\{l_{ik}\}_{k=1}^{K})=l_k)}{N_q}$. and the average accuracy is computed by $\texttt{Avg-Acc} = \frac{\sum_{i=1}^{N_q}\sum_{k=1}^{K} \mathbb{I}(l_{ik}=l_k)}{N_q K}$.

**Batch Diversity**    scRNA-seq datasets may contain cells from multiple experiment batches as only limited number of cells can be sequenced in one experiment. Ideally, the model should be agnostic to batch effects and datasets from different batches should mix well in the latent embedding space while preserving the biological information. In that case, the query cells can be linked with target cells from different biological studies to reveal common associations across cell states. Therefore, we designed a metric to quantify the batch diversity of the retrieved cells. Assume the batch labels of the retrieved cells are $\{\{b_{ik}\}_{k=1}^{K}\}_{i=1}^{N_q}$ and there are $M$ unique batch labels $\{b_i\}_{i=1}^{M}$. The batch diversity is defined as $\texttt{BatchDiv} = \frac{\sum_{i=1}^{N_q} \mathrm{Entropy}(\{\{b_{ik}\}_{k=1}^{K}\})}{N_q}$ and $\mathrm{Entropy}(\{b_{ik}\}_{k=1}^{K}) = \sum_{m=1}^{M} -\frac{\sum_{i=1}^{N_q} \mathbb{I}(b_{ik}=b_m)}{N_q} \log(\frac{\sum_{i=1}^{N_q} \mathbb{I}(b_{ik}=b_m)}{N_q})$. Intuitively, if batch diversity is higher, the model can better mix up the datasets from different batches. Different from kBET (Büttner et al., 2019) that measures the batch mixing with the global cell embedding cluster , the batch diversity metric measures the batch mixing property for individual cells thus is more suitable for cell retrieval evaluation.

**Recall across Omics**    For single-cell multiomics datasets, there are paired scRNA-seq and scATAC-seq profiles for cells. Therefore, we compute the top $K$ recall of cross-omic retrieval. i.e., for query cell $\{x_i\}_{i=1}^{N_q}$, we retrieve cells $\{\{y_{ik}\}_{k=1}^{K}\}_{i=1}^{N_q}$ from the reference cells and compute the recall of retrieval using $\texttt{Recall}_k = \sum_{i=1}^{N_q}\sum_{k=1}^{K} \frac{\mathbb{I}(y_{ik}=y_i)}{N_q}$. Intutively, the higher $K$, the more likely the matched cells from another omic can be retrieved thus $\texttt{Recall}_k$ will be higher.

### 3.2.2 LABEL-FREE EVALUATION

A critical challenge in evaluating cell retrieval is the lack of ground-truth cell-to-cell relationship annotations. Unlike traditional information retrieval setting with ground-truth query-target pairs, there are no ground-truth cell pair annotations. Even though single-cell datasets are usually annotated with different cell types, using the cell type annotation accuracy as the sole evaluation metric can be biased. In many cases, the cell types are annotated based on cell clustering with low-dimensional representation (t-SNE, UMAP) of cells, therefore cannot be considered as fully accurate (Heimberg et al., 2023). Inspired by the voting theory (O'Connor & Robertson, 2003) that performs a systematic aggregation of results in order to achieve consensus, we hypothesis that reference cells that are commonly retrieved by different single-cell retrieval methods are likely to be more biologically relevant with the query cells. We measure the similarity between different levels of consistency between the results from different single-cell retrieval methods.

**Consistency between the retrieved cells** Following Fan et al. (2024a), we propose to compute the average overlap between the retrieved cells from each method for the same query cells. Given query cell $x_i$, retrieval methods $a$ and $b$ may retrieve cells $\{y_{ik}^a\}$ and $\{y_{ik}^b\}$ respectively. We compute the Jaccard Similarity between $\{y_{ik}^a\}_{k=1}^K$ and $\{y_{ik}^b\}_{k=1}^K$ and average over all query cells as the `AvgOverlap` score.

**Consistency between the DE genes from the retrieved cells** Simply comparing the overlap of retrieved cells ignores the gene expression similarity between retrieved cells. Therefore, we analyze the consistency of gene expression features and patterns of the retrieved cells from different methods by comparing their DE (Differentially Expressed) genes. For each query cell and its retrieved cells, we computed their DE genes compared with all other cells. The DE genes are defined as genes with statistically higher expression compared with background (Wilcoxon rank-sum test, $p$-value $< 0.02$ and log-fold changes $> 0.5$). Then, we computed the Jaccard similarity between the DE genes across all query cells. Therefore, this method better captures the consistency between retrieved cells by considering their DE gene patterns.

### 3.3 Evaluation Settings

#### 3.3.1 Implementation

Different single-cell retrieval methods have different cell embedding and cell retrieval approaches. From the cell embedding perspective, except for CellFishing.jl, all other methods perform retrieval based on low-dimensional cell embedding. For VAE-based method, we train the corresponding method using the reference cells $\{y_i\}_{i=1}^{N_r}$ and extract the low-dimensional embedding for both $\{x_i\}_{i=1}^{N_q}$ and $\{y_i\}_{i=1}^{N_r}$ ($\{\text{VAE-Enc}(x_i)\}_{i=1}^{N_q}$ and $\{\text{VAE-Enc}(y_i)\}_{i=1}^{N_r}$). For scFM-based method, the foundation models are used in a zero-shot manner without additional tuning to avoid bias, we encode $\{x_i\}_{i=1}^{N_q}$ and $\{y_i\}_{i=1}^{N_r}$ into low-dimensional embedding ($\{\text{FM-Enc}(x_i)\}_{i=1}^{N_q}$ and $\{\text{FM-Enc}(y_i)\}_{i=1}^{N_r}$).

From the cell retrieval perspective, CellFishing.jl uses local senstive hashing (LSH) to directly search using the gene count vector. For all other methods with dense low-dimensional cell embeddings, we implement the retrieval framework using the widely used dense vector search tool Faiss (Johnson et al., 2019). Details regarding the implementation can be found in our codebase.

#### 3.3.2 Evaluation Datasets

We utilized multiple commonly used scRNA-seq datasets to evaluate the effectiveness of cell retrieval. These datasets span multiple species, tissues and experiment platforms.

**Multi-platform evaluation** Multiple scRNA-seq technologies have been developed with different tagging methods and sequencing libraries, thus different sequencing technologies may exhibit significant technical variation. With the increasing number of sequencing technologies, it is vital to evaluate whether single-cell retrieval methods could retrieve cells from different platforms with high accuracy. Therefore, we adopted two scRNA-seq datasets including multiple platforms, the PBMC dataset spanning 9 different sequencing platforms (Ding et al., 2020) and the human pancreas dataset spanning 4 different sequencing platforms (Luecken et al., 2022).

**Multi-species evaluation** Single-cell sequencing has been carried out extensively in different species. Analysis of single-cell datasets from diverse organisms is vital for understanding evolutionary processes of conservation and diversification of cell types. Therefore, we evaluated the cross-species retrieval capabilities of single-cell retrieval methods using two large scale human (Consortium* et al., 2022) and mouse (Consortium, 2020) single-cell atlas spanning more than 10 tissues.

**Multi-omics evaluation** Single-cell multi-omics profiling allows for measurements of transcriptome (scRNA-seq) and chromatin accessibility (scATAC-seq) for the same cells, i.e. $\{x_i\}_{i=1}^N$ (scRNA-seq) and $\{y_i\}_{i=1}^N$ (scATAC-seq) datasets where $x_j$ and $y_j$ measures the same cell $j$. We collected three widely used single-cell multiomics profiling datasets, namely 10X PBMC (pbm, 2020), Chen-2019 (Chen et al., 2019) and Ma-2020 (Ma et al., 2020). These datasets contain 9631, 9190 and 32231 cells respectively.

Table 1: Evaluating single-cell FMs in cross-platform retrieval setting on human PBMC dataset (`Vote-Acc` and `BatchDiv` ). $K$ denotes the number of cells retrieved given 1 query cell. Setting: Leave-one-out. Given scRNA-seq sequencing results from $N$ platforms, the sequencing results from $N-1$ platforms are used as reference and the remaining platform is used as query. Bold numbers, underline numbers, and dashed numbers show the first, second, and third best scores respectively.

| K | 1 | | 5 | | 10 | | 20 | | 50 | | 100 | |
|---|---|---|---|---|---|---|---|---|---|---|---|---|
| Metric | Vote Acc | Batch Div | Vote Acc | Batch Div | Vote Acc | Batch Div | Vote Acc | Batch Div | Vote Acc | Batch Div | Vote Acc | Batch Div |
| PCA | 0.556 | 0.000 | 0.592 | 0.513 | 0.604 | 0.620 | 0.613 | 0.702 | 0.604 | 0.796 | 0.583 | 0.861 |
| CellFishing.jl | 0.812 | 0.000 | 0.843 | 0.389 | 0.851 | 0.483 | 0.858 | 0.560 | 0.861 | 0.653 | 0.859 | 0.733 |
| scVI | 0.778 | 0.000 | 0.810 | 0.252 | 0.822 | 0.316 | 0.832 | 0.374 | 0.830 | 0.453 | 0.830 | 0.531 |
| LDVAE | 0.754 | 0.000 | 0.800 | 0.334 | 0.808 | 0.406 | 0.812 | 0.461 | 0.815 | 0.523 | 0.810 | 0.579 |
| CellBlast | 0.762 | 0.000 | 0.802 | 0.328 | 0.812 | 0.407 | 0.817 | 0.472 | 0.812 | 0.545 | 0.812 | 0.599 |
| scFoundation | 0.849 | 0.000 | 0.876 | 0.324 | 0.882 | 0.399 | 0.884 | 0.463 | 0.887 | 0.555 | **0.889** | 0.639 |
| scGPT | 0.803 | 0.000 | 0.838 | 0.349 | 0.844 | 0.430 | 0.849 | 0.497 | 0.844 | 0.582 | 0.840 | 0.654 |
| SCimilarity | 0.850 | 0.000 | 0.876 | 0.539 | 0.883 | 0.681 | 0.885 | 0.794 | 0.886 | 0.931 | 0.884 | 1.044 |
| UCE | **0.852** | 0.000 | **0.878** | 0.471 | **0.884** | 0.600 | **0.886** | 0.706 | **0.888** | 0.824 | 0.886 | 0.913 |
| Geneformer | 0.768 | 0.000 | 0.810 | 0.526 | 0.824 | 0.662 | 0.828 | 0.769 | 0.820 | 0.893 | 0.810 | 0.994 |
| scMulan | 0.822 | 0.000 | 0.857 | 0.487 | 0.863 | 0.609 | 0.863 | 0.706 | 0.858 | 0.814 | 0.854 | 0.893 |
| CellPLM | 0.806 | 0.000 | 0.841 | 0.413 | 0.852 | 0.524 | 0.858 | 0.615 | 0.861 | 0.719 | 0.860 | 0.792 |

### 3.3.3 DATASET PRE-PROCESSING

**General preprocessing** Different single-cell retrieval methods exhibit different pre-processing steps. For example, scFoundation does not select highly variable genes while scGPT selects the top 4000 highly variable genes. To avoid bias, we adopt the default pre-processing method of each method respectively.

**Cross-species mapping** To perform cross-species retrieval, gene alignment is crucial as human and mouse genes are different. Mouse and human have well annotated one-to-one gene homolog mapping as they have close evolutionary distance and have been well studied. We download human and mouse gene homology from existing database[2] and align the mouse and human scRNA-seq datasets.

**Cross-omic mapping** As scFMs are pre-trained on scRNA-seq datasets, directly applying them to other omics such as scATAC-seq is not trivial. For each dataset, We retained 80000 highly variable peaks for scATAC-seq and 8000 highly variable genes for scRNA-seq. We followed the standard implementation from DeepMAPS (Ma et al., 2023a) to align scATAC-seq peaks to the same gene space as scRNA-seq based on gene regulatory potential.

## 4 RESULTS

### 4.1 CROSS PLATFORM RETRIEVAL

We first benchmarked the performance of single-cell retrieval methods across different platforms using the human PBMC (Table 1) and human pancreas (Appendix Table 1) datasets. The detailed results for each platform can be found in Appendix.

**Benchmarking on Human PBMC dataset** On the human PBMC dataset (Table 1), UCE, scFoundation and SCimilarity show substantial advantage over all other methods. Meanwhile, we also noticed that the batch diversity of retrieved cells from scFMs is also significantly higher than other methods, indicating these methods can better find cells across different experiment platforms.

**Benchmarking on Human pancreas dataset** On the human pancreas dataset (Appendix Table 1), scFMs do not show significant advantage in cell type vote accuracy, while CellFishing.jl has promising performance. VAE-based methods LDVAE and scVI also perform quite well in this setting.

---

[2] `https://www.informatics.jax.org/downloads/reports/HOM_MouseHumanSequence.rpt`

Among all scFMs, SCimilarity still shows the best performance considering the cell type vote accuracy and batch diversity. As the human PBMC scRNA-seq datasets are much more accessible compared with the human pancreas datasets, the performance gap can be attributed to the pre-training data distribution difference.

## 4.2 Cross Species Retrieval

Table 2: Evaluating single-cell FMs in cross-species retrieval setting (`Vote-Acc`). $K$ denotes the number of cells retrieved given 1 query cell. Bold numbers, underlined numbers, and dashed numbers show the first, second, and third best scores respectively.

| Settings | Mouse->Human | | | | | | | Human->Mouse | | | | | | |
|---|---|---|---|---|---|---|---|---|---|---|---|---|---|---|
| K | 1 | 5 | 10 | 20 | 50 | 100 | Avg | 1 | 5 | 10 | 20 | 50 | 100 | Avg |
| PCA | 0.690 | 0.692 | 0.696 | 0.695 | 0.682 | 0.668 | 0.687 | 0.799 | 0.830 | 0.843 | 0.835 | 0.802 | 0.794 | 0.817 |
| CellFishing.jl | 0.687 | 0.701 | 0.706 | 0.706 | 0.699 | 0.686 | 0.698 | 0.749 | 0.764 | 0.760 | 0.739 | 0.719 | 0.717 | 0.741 |
| scVI | 0.709 | 0.714 | 0.714 | 0.702 | 0.634 | 0.643 | 0.686 | 0.769 | 0.798 | 0.790 | 0.789 | 0.786 | 0.772 | 0.784 |
| LDVAE | 0.671 | 0.680 | 0.693 | 0.696 | 0.690 | 0.680 | 0.685 | 0.768 | 0.782 | 0.788 | 0.778 | 0.789 | 0.793 | 0.783 |
| CellBlast | 0.534 | 0.555 | 0.522 | 0.518 | 0.503 | 0.503 | 0.523 | 0.583 | 0.606 | 0.603 | 0.607 | 0.610 | 0.595 | 0.601 |
| scFoundation | 0.748 | 0.754 | 0.730 | 0.726 | 0.734 | 0.745 | 0.740 | 0.866 | 0.859 | 0.863 | 0.852 | 0.850 | 0.846 | 0.856 |
| scGPT | 0.759 | 0.760 | 0.763 | 0.766 | 0.766 | 0.761 | 0.763 | 0.841 | 0.846 | 0.845 | 0.849 | 0.810 | 0.796 | 0.831 |
| SCimilarity | 0.799 | 0.796 | 0.804 | 0.779 | 0.794 | 0.798 | **0.795** | 0.897 | 0.911 | 0.911 | 0.914 | 0.883 | 0.881 | **0.900** |
| UCE | 0.785 | 0.785 | 0.776 | 0.780 | 0.799 | 0.801 | 0.788 | 0.882 | 0.905 | 0.906 | 0.901 | 0.874 | 0.860 | 0.888 |
| Geneformer | 0.594 | 0.595 | 0.597 | 0.588 | 0.568 | 0.537 | 0.580 | 0.608 | 0.619 | 0.615 | 0.611 | 0.611 | 0.604 | 0.611 |
| scMulan | 0.766 | 0.772 | 0.770 | 0.721 | 0.699 | 0.733 | 0.744 | 0.834 | 0.852 | 0.858 | 0.862 | 0.836 | 0.834 | 0.846 |
| CellPLM | 0.731 | 0.741 | 0.740 | 0.743 | 0.746 | 0.747 | 0.741 | 0.859 | 0.862 | 0.866 | 0.858 | 0.839 | 0.832 | 0.853 |

We then evaluated the single-cell retrieval methods in cross-species retrieval setting. We benchmarked the model performance using human and mouse scRNA-seq datasets across 10 tissues.

**Superiority of scFMs over other methods** In the challenging cross-species setting, scFMs significantly outperform traditional methods and VAE-based methods. The results are shown in Table 2. For example, on the mouse to human retrieval setting, the vote-acc of the best non-scFM method CellFishing.jl is 10% lower than the best scFM method SCimilarity.

**Comparison between multi-species scFM and human-centered scFM** UCE is the only scFM pretrained with multi-species datasets, while other scFMs are human-centered with only human scRNA-seq pre-training datasets. As shown in Table 2, UCE ranks second among all single-cell retrieval methods, but does not show significant improvement over human-centered scFMs. It indicates that human-centered scFMs can well generalize in human-mouse cross-species retrieval setting even without explicitly trained on mouse datasets. It is also important to notice that human and mouse have explicit one-to-one gene homolog thus human-centered scFMs can be directly applied to cross-species retrieval. Generalization of scFMs to other distant species without explicit homolog mapping still remains an open problem.

## 4.3 Cross Omic Retrieval

Single-cell sequencing technologies measure individual cell state from different omics, thus whether the single-cell retrieval methods can find relevant cells spanning different omics is an important evaluation of cell retrieval capabilities. In additional to cell type retrieval accuracy, we also utilized the recall across omic metric to test whether the retrieval methods can find the exact match cells across omics. The results are shown in Table 3. We evaluated the model performance on three widely studied single-cell multiomics datasets spanning different tissues and species.

**Advantage of scFMs on human multi-omics datasets** UCE, scFoundation and SCimilarity perform best on the 10x Multiome datasets and show significant advantages compared with other non-scFM methods in both retrieval directions.

**Poor performance of scFMs on mouse multi-omics datasets** On the Chen-2019 and Ma-2020 dataset, scFM methods all perform poorly and VAE-based methods including scVI, LDVAE and

Table 3: Evaluating single-cell FMs in cross-omic retrieval setting. We set the retrieval cell number for each query cell as 50. Bold numbers, underline numbers, and dashed numbers show the first, second, and third best scores respectively.

| | 10x Multiome (Human Blood) | | | | Chen-2019 (Mouse Cortex) | | | | Ma-2020 (Mouse Skin) | | | |
| --- | --- | --- | --- | --- | --- | --- | --- | --- | --- | --- | --- | --- |
| Setting | scRNA->scATAC | | scATAC->scRNA | | scRNA->scATAC | | scATAC->scRNA | | scRNA->scATAC | | scATAC->scRNA | |
| Method | Recall | Vote-Acc | Recall | Vote-Acc | Recall | Vote-Acc | Recall | Vote-Acc | Recall | Vote-Acc | Recall | Vote-Acc |
| PCA | 0.006 | 0.225 | 0.048 | 0.522 | 0.008 | 0.214 | 0.012 | 0.222 | 0.008 | 0.392 | 0.003 | 0.282 |
| CellFishing.jl | 0.027 | 0.613 | 0.063 | 0.622 | 0.009 | 0.333 | 0.020 | **0.284** | 0.010 | 0.384 | 0.034 | 0.412 |
| scVI | 0.009 | 0.204 | 0.006 | 0.212 | 0.016 | 0.371 | 0.012 | 0.231 | 0.031 | 0.677 | 0.016 | 0.414 |
| LDVAE | 0.010 | 0.233 | 0.006 | 0.187 | 0.021 | **0.472** | 0.015 | 0.283 | 0.034 | **0.696** | 0.026 | **0.506** |
| CellBlast | 0.083 | 0.580 | 0.075 | 0.658 | 0.016 | 0.266 | 0.025 | 0.263 | 0.019 | 0.596 | 0.021 | 0.499 |
| scFoundation | 0.105 | 0.808 | 0.080 | 0.691 | 0.008 | 0.232 | 0.006 | 0.199 | 0.014 | 0.379 | 0.011 | 0.296 |
| scGPT | 0.055 | 0.673 | 0.034 | 0.550 | 0.009 | 0.263 | 0.009 | 0.221 | 0.009 | 0.325 | 0.005 | 0.227 |
| SCimilarity | 0.077 | 0.726 | 0.053 | 0.535 | 0.013 | 0.279 | 0.006 | 0.188 | 0.013 | 0.362 | 0.011 | 0.308 |
| UCE | 0.099 | **0.855** | 0.078 | **0.709** | 0.011 | 0.272 | 0.006 | 0.203 | 0.014 | 0.390 | 0.008 | 0.274 |
| Geneformer | 0.013 | 0.320 | 0.006 | 0.196 | 0.006 | 0.243 | 0.008 | 0.240 | 0.002 | 0.162 | 0.002 | 0.196 |
| scMulan | 0.044 | 0.639 | 0.036 | 0.556 | 0.011 | 0.292 | 0.009 | 0.239 | 0.009 | 0.325 | 0.006 | 0.261 |
| CellPLM | 0.067 | 0.701 | 0.063 | 0.541 | 0.012 | 0.344 | 0.010 | 0.197 | 0.012 | 0.380 | 0.010 | 0.292 |

CellBlast achieve the best performance. The performance gap across the human and mouse multi-omics datasets could be attributed to the large gap between the pre-training corpus of scFMs (human scRNA-seq) and the testing omic and species (mouse scATAC-seq). scFMs can generalize well either on another close species (mouse scRNA-seq) or another omic (human scATAC-seq), but on mouse scATAC-seq which is quite distant from the pre-training corpus, scFMs would be likely to fail. Even for UCE which is pre-trained on multi-species datasets, the performance is still relatively poor. Therefore, when the target species and omics are both distant from the pre-training corpus, single-cell FMs may not be preferred in cell retrieval.

**Poor performance of single-cell retrieval methods to find exact match cells** The recall metric measures whether single-cell retrieval methods can identify the exact match cells using information from another omic. The result shows that even all methods perform better than random, most methods do not show significant improvement over the random baseline, highlighting that identifying the matching cells aross omic is still a highly challenging task.

## 4.4 LABEL-FREE EVALUATION OF SINGLE-CELL RETRIEVAL METHODS

Cell type annotations only provide coarse grained information as cells in the same cell type may belong to different regions or different stages. Meanwhile, cell type annotations can be highly inconsistent across different studies and can even sometimes be wrong. Motivated by this, besides computing the agreement between cell type annotations and retrieval results, comparing the consistency of the retrieval cells between different methods may serve as another signal of correctness for cell retrieval. As mentioned in Section 3.2, we proposed the label-free evaluation of single-cell retrieval methods, considering both the consistency between the retrieved cells and the consistency between the DE patterns from the retrieved cells.

First, we evaluated the overlap between the retrieved cells between different single-cell retrieval methods. As shown in Fig. 2a, we found that Cellfishing.jl, scFoundation and scGPT have high overlap on the retrieved cells. Meanwhile, UCE and SCimilarity also have high overlap with these methods. In Fig. 2b, we visualized the `AvgOverlap` score of single-cell retrieval methods and found that scFoundation, CellFishing.jl and scGPT rank the highest among all methods. It is vital to validate that whether the `AvgOverlap` score correlates with the `Vote-Acc` metric so that `AvgOverlap` can be a signal of correctness of cell retrieval when no high-quality cell type annotations are available. As shown in Fig. 2c, the label-free metric `AvgOverlap` and label-dependent metric `Vote-Acc` are strongly correlated across four benchmarking datasets. Therefore, we could use the label-free metric `AvgOverlap` to evaluate single-cell retrieval methods even when the cell type annotations on the target dataset are not available.

Second, we further computed the consistency of differential gene expression patterns from the retrieved cells of different methods. Simply relying on the overlap between cell indexes can be biased as it ignores the semantic information from cells, i.e. the gene expression patterns. We analyzed whether the retrieved cells given the same query cell from each method have consistent differential

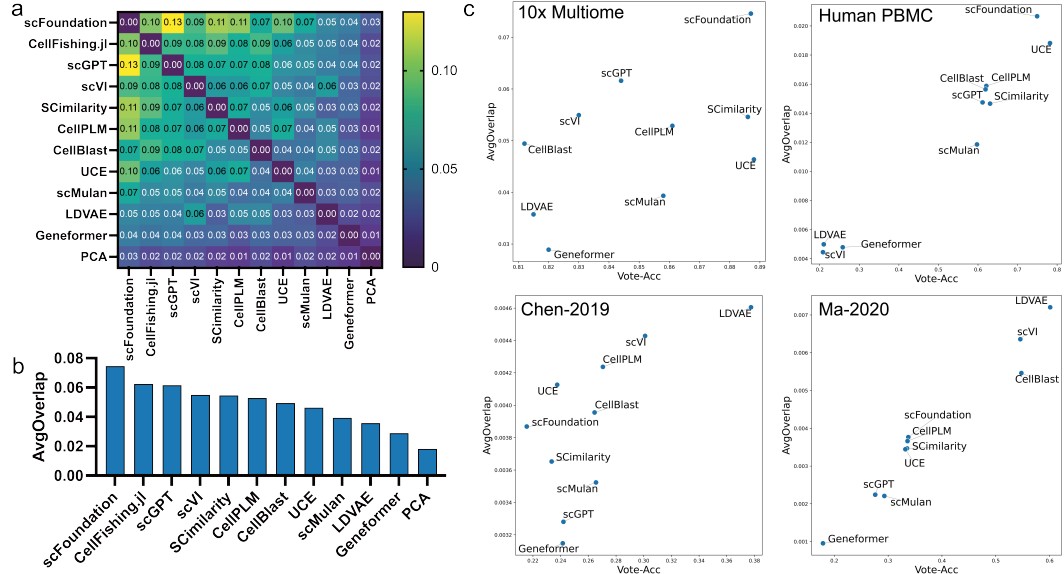

Figure 2: Consistency between the retrieved cells from different methods. a. Heatmap of the overlap between the retrieved cells between different methods on Human PBMC dataset (K=50). b. `AvgOverlap` score of 12 single-cell retrieval methods on Human PBMC dataset (K=50). c. Correlation between `Vote-Acc` and `AvgOverlap` on four benchmarking datasets across different methods.

gene expression patterns. Following the steps in Section 3.2.2, we visualized the Jaccard similarity of identified DE genes between all query cells on human PBMC dataset for CD4+ T cells in Fig. 3a for different methods. As we can see, some single-cell retrieval methods such as scVI and LDVAE do not show significant variation within the cell type, while the retrieved cells from scFMs and CellFishing.jl have distinct differences and sub-groups with similar DE patterns can be identified within a cell type (red boxes). We can see that the DE gene patterns of SCimilarity are very similar to that of Cellfishing, scFoundation and scGPT while scVI does not exhibit similar patterns, which indicates that these top scFMs can identify common cells with similar gene expression patterns and the sub-groups they find may correspond to certain unannotated sub-types of CD4+ T cells. Besides visual inspection, we next quantitatively evaluated how similar are sub-groups identified by different methods in Fig. 3b. Concretely, we computed the Jaccard similarity between identified DE genes from different methods for each query cell and average over all the query cells. Intuitively, the higher average similarity between methods means that the sub-groups they identified have more similarity. We found that top cell retrieval methods in quantitative benchmarking, including scGPT, SCimilarity, UCE, scFoundation and CellFishing.jl, also have higher similarity in cell DE sub-groups. The cell DE sub-groups identified in common can be further explored and explained by biologists.

## 5 RELATED WORKS

### 5.1 SINGLE-CELL FMS BENCHMARKING

There are various attempts in evaluating single-cell FMs from multiple aspects. For example, Alsabbagh et al. (2023) mainly evaluate the cell type annotation capabilities of scFMs. Kedzierska et al. (2023) focus on evaluating the zero-shot cell embedding capabilities of single-cell FMs. Zhao et al. (2023) evaluates single-cell FMs in terms of cell annotation, gene annotation, perturbation response and imputation. With the increasing number of single-cell FMs, evaluating and benchmarking these FMs have been considered of greater significance.

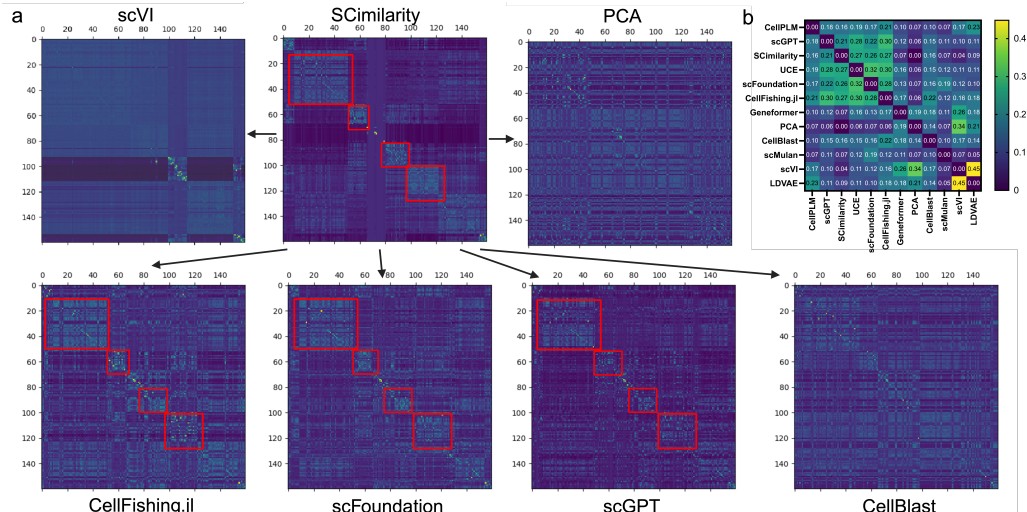

Figure 3: Consistency between the DE genes of the retrieves cells from different methods. (a). Heatmap showing the Jaccard similarity between DE genes of query cells. Each entry $(i, j)$ in the heatmap indicates the Jaccard similarity of the DE genes computed from the query cell $i$ and its corresponding retrieved cells and the DE genes computed from the query cell $j$ and its corresponding retrieved cells. The red boxes indicates the sub-group of cells that share similar DE gene patterns. (b). Heatmap showing the overlap of DE genes from the retrieved cells across different methods.

## 5.2 BIOLOGICAL DATA SEARCH AND RETRIEVAL

Retrieving cells from large scale biological databases with machine learning methods has already been widely studied across different biological domains. For example, in the protein domain, Hong et al. (2024) searches large scale protein databases with pre-trained protein large language model, Ma et al. (2023b) jointly trains the protein retriever and protein language model to train protein language models with stronger representation abilities. In the neuron domain, Fan et al. (2024b) pre-trains a foundation model to retrieve similar neurons.

## 6 CONCLUSION AND LIMITATION

In this paper, we comprehensively benchmarked and evaluated 12 existing single-cell retrieval methods. We proposed two types of evaluation metrics: label-free evaluation and label-dependent evaluation to assess the capabilities of single-cell retrieval methods. The key recommendations are summarized as follows: scFMs show promising performance in retrieving similar cell states given query cells, but are not reliable when the target species and omics are distant from the pre-training dataset; Traditional non-machine learning cell retrieval methods still yield promising results in major settings, which should be used as strong baselines in further method development; In cases where the target species and omics are distant from the scFM pre-training datasets, VAE-based methods such as scVI and LDVAE are good alternatives; Label-free metrics yield consistent results as label-dependent results, thus can be adapted especially when the cell type labels are inconsistent across studies. We envision the development of large-scale scFMs pre-trained on more species, tissues, omics that enables the development of a general cell "search engine".

**Limitations**: Quantitatively benchmarking of single-cell retrieval methods still heavily relies on the cell type annotations. However, these annotations may be inconsistent across studies or incorrect, therefore causing bias in the evaluation outcome. In this paper, we aim to alleviate this issue by proposing a label-free evaluation methodology. We believe that more label-free validation methods of cell retrieval performance are key to evaluating the single-cell retrieval methods at scale across more diverse datasets.

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
