# OpenReview forum: "Evaluating Single-Cell Foundation Models for Cell Retrieval"
_ICLR.cc/2025/Conference — ICLR 2025 Conference Withdrawn Submission_

### Official Review · Reviewer_6nWW · 2024-11-01

**Soundness:** 1
**Presentation:** 2
**Contribution:** 3
**Rating:** 6
**Confidence:** 5

**Summary:**

This paper presents a comprehensive benchmark for evaluating single-cell RNA sequencing (scRNA-seq) retrieval methods, including single-cell foundation models (scFMs), VAE-based models, and traditional approaches. Addressing challenges in cell retrieval across diverse datasets, the authors introduce label-dependent and label-free metrics to benchmark 12 retrieval methods across cross-platform, cross-species, and cross-omics scenarios. This work contributes an important foundation for future development and evaluation of single-cell retrieval methods.

**Strengths:**

(1)	This paper addresses a critical problem in single-cell data analysis by introducing a standardized benchmarking framework for single-cell RNA-seq retrieval methods.
(2)	The paper presents a well-structured and comprehensive evaluation across 12 retrieval methods, spanning non-machine learning, VAE-based, and scFM-based approaches.  The authors carefully selected diverse datasets (cross-platform, cross-species, and cross-omics) to ensure robust and unbiased comparisons.

**Weaknesses:**

(1)	This papers primarily evaluates non-machine learning, VAE-based, and scFM-based methods, overlooking other potentially relevant  graph neural networks (GNNs) based methods, such as scGNN (Nature Communications2021) and scMoGNN (KDD2022). In addition, more foundation models should also be considered, such as scBERT (Nature Machine Intelligence2022), scCLIP (NeurIPS 2023) and tGPT (iScience2023).
(2)	There are no appropriate statistical tests for the results in Table 1~Table 3, although measures such as variance or standard deviation would provide a clearer understanding of the relative strengths and robustness of each method.
(3)	Although the paper mentions the limitations of the scFMs methods in cross-species and cross-omics data, it does not deeply analyze why these models perform poorly in specific situations and how to improve them. For example, can the batch effect of the data be eliminated before pre-training or cell retrieval? In addition, can other multi-omics pre-processing methods and alignment tools Seurat, ArchR and SnapATAC be considered instead of DeepMAPS?

**Questions:**

(1)	This paper proposes some new evaluation metrics, but does not fully explain the biological significance of these metrics, such as batch diversity. In actual situations, does batch diversity explain the consistency of cell types or states in different experimental batches? Besides, does small batches of cells or uneven distribution affect the batch diversity?
(2)	The multi-omics evaluation section needs to be further improved, because it currently only involves two omics, scRNA-seq and scATAC-seq. It is recommended that the authors add further evaluation of DNA methylation data and single-cell protein data. In addition, how to achieve cell retrieval in unpaired single-cell multi-omics data? A potential method is to build associations between different omics data through prior knowledge and achieve alignment of unmatched data for cell retrieval (GLUE, Nature Biotechnology2022).
(3)	In section 4.2, why are scFMs superior to other methods? In Fig. 2c, How to calculate the correlation between two metrics?
(4)	This paper summarizes the advantages and disadvantages of different methods, but does not provide detailed suggestions on how to improve these methods. This makes the benchmark less practical and fails to provide clearer improvement directions for future research, such as how to improve the model's ability to generalize across species or across omics.

---

### Official Review · Reviewer_9zAf · 2024-11-03

**Soundness:** 2
**Presentation:** 2
**Contribution:** 2
**Rating:** 3
**Confidence:** 3

**Summary:**

The paper presents a comprehensive benchmark for evaluating the efficiency and accuracy of single-cell retrieval methods using single-cell RNA-seq databases. It introduces a diverse set of benchmarks encompassing various models and methods with a special emphasis on single-cell foundation models (scFMs). The evaluation spans across label-dependent and label-free metrics, offering insights into the methods' performance in different settings, such as cross-platform, cross-species, and cross-omics comparisons.

**Strengths:**

1. The development of a benchmark for evaluating cell retrieval methods is a notable effort, addressing the current gap in the literature.
2. The comprehensive comparison across various metrics and conditions provides a good resource for the community, potentially guiding future developments in the field.

**Weaknesses:**

1. The presentation of results is convoluted, with tables that are difficult to interpret, reducing the impact of the findings.
2. The paper does not adequately discuss the potential for foundation models to have encountered test data during training, which is crucial for interpreting their performance accurately.

**Questions:**

1. (Line 181) there is a typo for avg-acc: l_i instead of l_k
2. Why using the same metrics but named differently for different omics: "recall" and "avg-acc"?
3. Sec. 3.3.3 would "general processing" make results for different methods not optimal? Some methods may have this preference built in their algorithm
4. Table 1, 2 and 3 have abundent numbers. Could there be better and more friendly way to demonstrate the results?
5. There is a huge limitation: the paper does not discuss the relationship between the pre-training data for each FM and the downstream datasets. Is it possible that FMs perform well because they have seen these cells in the training data before?
6. In Sec. 4.3., is there any statistics evidence that human scRNA-seq, mouse scRNA-se and human scATAC-seq are similar, while are different from mouse scATAC-seq?

---

### Official Review · Reviewer_RRQw · 2024-11-04

**Soundness:** 3
**Presentation:** 3
**Contribution:** 3
**Rating:** 5
**Confidence:** 5

**Summary:**

The authors develop an evaluation system for the cell retrieval task in the field of single cell genomics.

**Strengths:**

Cell retrieval is an important challenge. This paper presents the first evaluation system for this task.

**Weaknesses:**

The evaluation system could be more comprehensive (e.g. how about looking at enriched pathways as well as genes?).

It is not clear how well batch effects are handled - it would be best to evaluate based on a data set that is heavily affected by batch effects.

There seems to be no claim to make the evaluation methods available for others to use as a software implementation, as is typically done for metrics in other fields (e.g. ROUGE and BLEU in NLP)

**Questions:**

How will others be able to perform this evaluation on new cell retrieval methods?

How well do the methods work on a strongly batch affected data set?

---

### Official Review · Reviewer_DDdS · 2024-11-04

**Soundness:** 3
**Presentation:** 3
**Contribution:** 3
**Rating:** 8
**Confidence:** 3

**Summary:**

This paper benchmarks the existing single-cell foundation models on the cell retrieval task to alleviate the lack of standard evaluation metrics and comprehensive benchmark datasets. The overall evaluation benchmark is relatively comprehensive, including 12 existing single-cell retrieval methods, with 2 different evaluation metrics and 3 different settings. Additionally, some notable findings and limitations are discussed. Overall, this is a good benchmark paper that will contribute to the development of the single-cell retrieval field.

**Strengths:**

The benchmark setting is very comprehensive, covering three major categories of single-cell retrieval methods (non-machine learning methods, VAE-based methods, and single-cell basic model-based methods), and two evaluation criteria including label-dependent evaluation (cell type accuracy, batch diversity, and cross-omics recall) and label-free evaluation due to the lack of ground-truth. Furthermore, in the experimental setup, diverse settings (cross-platform, cross-species, and cross-omics) were considered. This article is well organized and very easy to understand. The source code is also uploaded in the supplementary material, which has a clear structure and the results are easy to reproduce.

**Weaknesses:**

These are not weaknesses, just some suggestions that might be helpful.
1. In total, 12 methods were collected and divided into 3 groups.It would be better if the differences and characteristics of these methods could be explained in more detail. This will be more helpful for researchers who want to start joining this field to understand these models more deeply and run some baselines/benchmarks.
2. It would be more convincing if the authors could add more details about the datasets they used in the paper. Also, add some scripts to your source code repository and include some preprocessing details (maybe I overlooked this part in your code).
3. In the source code repository, perhaps some more detailed documentation is needed to help researchers integrate your code to do some further experiments.

**Questions:**

The current manuscript and uploaded source code are both very good and well structured. However, single cells are advancing rapidly and many new models/algorithms are being developed. I would like to ask whether the emerging models are easy to incorporate into your benchmark setup?

---

### Official Review · Reviewer_s3Gw · 2024-11-04

**Soundness:** 2
**Presentation:** 3
**Contribution:** 2
**Rating:** 5
**Confidence:** 3

**Summary:**

The authors proposed a comprehensive evaluation benchmark to assess the capabilities of 12 existing single-cell retrieval methods from three classes: non-machine learning method, VAE-based methods and single-cell foundation model (scFM) based methods. They proposed a series of label-dependent and label-free evaluation metrics to assess the performance of single-cell retrieval methods.
They found that scFMs show good performance in retrieving similar cell states for a given query cell, but when the target species and omics are far away from the pre-training dataset, VAE-based methods such as scVI and LDVAE are better choices. Their work reveals the challenges and shortcomings of the current single-cell retrieval methods, forming a basis for the future enhancement of single-cell retrieval strategies.

**Strengths:**

1. The paper conducts a systematic benchmark of 12 single-cell retrieval methods, including non-machine learning methods, variational autoencoder (VAE)-based methods, and single-cell foundation model (scFM) methods. Through cross-platform, cross-species, and cross-omics settings, it provides a detailed evaluation of each method’s performance.

2. This study designs both label-dependent and label-independent evaluation metrics. Particularly, the label-independent metrics offer reliable assessment results in cases of missing or inconsistent labels, which serves as a strong complement to traditional evaluation methods.

3. The paper analyzes the generalization ability of models across various platforms and species, demonstrating the superiority of scFM in cross-platform retrieval and highlighting its limitations in cross-species retrieval.

**Weaknesses:**

1.	The paper did not conduct more experiments on the preprocessing stage, such as gene selection, which may have an impact on the retrieval effect of the model.

2.	In the “CROSS PLATFORM RETRIEVAL” part, only human data was used. Is it possible to add the evaluation results of the mouse cross-platform dataset?

3.	In the “CROSS OMIC RETRIEVAL” part, only chromatin accessibility was used. Can other omics be used to conduct experiments? We are not sure whether the differences in the results obtained from multi-omics data are due to the specificity of scATAC-seq data or the performance of the retrieval method itself.

4.	The authors lack a detailed description of the datasets, such as the size of each dataset and the size of the test/train sets (if applicable).

5.	The authors did not clearly specify the foundational database used for retrieval. If only test data is used as the database, this differs from real-world application scenarios.

6.	In the results section, only the proposed vote-acc was calculated without comparing it to avg-acc, making it difficult to demonstrate that this is a better metric.

**Questions:**

1.	More experiments should be done on the preprocessing part to explore how big the impact of this module is, such as selecting the same number of genes for experiments with different methods.

2.	Based on these experiments in the paper alone, the conclusion ”when the target species and omics are both distant from the pre-training corpus, single-cell FMs may not be preferred in cell retrieval” cannot be fully verified. If you can add cross-platform experiments on mouse data, as well as experiments on other omics data in the “CROSS OMIC RETRIEVAL” part, such as proteomics, the verification of this conclusion will be much stronger.

3.	The authors only considered adding BatchDiv as a metric to assess batch effects. Could further standardization methods be proposed to eliminate these differences and achieve better evaluation performance?

4.	The authors said “Top scFMs such as SCimilarity, UCE and scFoundation show significant advantage over other baseline methods in a zero-shot manner in the majority of settings, but still perform poorly when used on datasets distant from the pre-training database”. However, could this be due to an unreasonable dataset setting? Does a high proportion of long-tail data affect the fairness of the benchmark?

---

### Official Review · Reviewer_awRf · 2024-11-06

**Soundness:** 2
**Presentation:** 2
**Contribution:** 2
**Rating:** 3
**Confidence:** 4

**Summary:**

Given the increasing number of methods to query cells against existing databases and the lack of benchmarks, this paper presents multiple metrics to assess cell-retrieval performance across 12 existing methods. First, they assess how well cells with the same labels as the query can be retrieved across different batches, species, and data types. Then they propose that retrieval similarities across methods can act as a label-free way to benchmark the performance and show that this approach achieves outcomes consistent with the label-based methods. Based on the comparisons, the paper suggests no clear winner and outlines the advantages of single-cell foundation and VAE-based models for different tasks.

**Strengths:**

The paper evaluated a diverse set of existing methods and proposed a label-free way to evaluate their relative performance, which could be widely applicable to cases without known labels.

Benchmarking single-cell retrieval methods is essential to inform researchers on methodological choices.

**Weaknesses:**

In a lot of cases, users would like to query/annotate a small set of cells generated from their experiment against large reference databases, and query cells that come from a new context. The paper didn’t clarify or assess these use cases.

The paper didn’t specify how hyperparameters are selected for each of the 12 methods. The choice of hyperparameters can be vital for a model’s performance. Thus, the conclusions of the paper are not convincing.

The evaluations lack a baseline performance to assess how much power each method gains compared to random retrieval.

The evaluation of the multi-omics setup is only focused on RNA & ATAC, and the matching relies on a particular peak-gene matching method. Furthermore, the paper didn’t explore other data types or multi-omics matching methods for cross-omics cell retrieval. These limits the application scope of the benchmarking paper.

In the DE-gene-based label-free evaluation (Figure 3), the Jaccard similarity calculation seems to be incorrect as the matrices are not symmetric.

Multiple hypothesis correction is lacking for the differential expression calculation.

In the key takeaways, it is unclear how “distant” is defined and how the conclusions are reached.

**Questions:**

How do different methods compare if the query cell set is small and has non-overlapping labels with the reference databases?

What is the baseline performance for each task?

How are the hyperparameters selected for each method? How do different methods compare if hyperparameters are tuned for each method?

Will the cross-omics performance change across different methods when mapping genes and peaks based on all features instead of focusing on highly variable ones?

What is the cell retrieval performance for cross-omics tasks when using alignment methods such as UnionCom, MMD-MA, and SCOT?

How do different method perform when testing on other omic types, such as CITE-seq?
The similarities in label-free evaluation appears quite low, and the result tend to be inconsistent across different datasets. How does the choice of p-value cutoffs and logFC affect the conclusion?
How are Jaccard similarity matrices calculated in Figure 3 and why are the matrices not symmetric?
Could you make a summary table of datasets used for evaluation, their properties such as number of query and reference cells, and where they are used for various downstream evaluation metrics?

---

### Official Review · Reviewer_7r7q · 2024-11-08

**Soundness:** 3
**Presentation:** 3
**Contribution:** 2
**Rating:** 5
**Confidence:** 4

**Summary:**

This paper presents a comprehensive evaluation of 12 single-cell retrieval methods, including traditional non-machine learning approaches, VAE-based methods, and single-cell foundation models (scFMs). The authors propose both label-dependent and label-free evaluation metrics and conduct experiments across cross-platform, cross-species, and cross-omics settings. Through extensive benchmarking, they demonstrate the advantages of scFMs in most scenarios while highlighting the competitive performance of traditional methods, and validate the effectiveness of label-free evaluation metrics as alternatives when cell annotations are inconsistent.

**Strengths:**

1. The paper establishes a comprehensive benchmark by evaluating cell retrieval methods across multiple dimensions (cross-platform, cross-species, and cross-omics). The evaluation metrics cover both label-dependent aspects (cell type accuracy, batch diversity) and label-free measures (retrieval consistency, DE gene patterns), providing a balanced assessment approach.
2. The paper reveals several important insights: (a) while scFMs generally excel, they show limitations on datasets distant from their pre-training distribution; (b) traditional non-machine learning methods remain surprisingly competitive, challenging the assumption that complex models are always superior; (c) label-free metrics correlate well with label-dependent ones, offering alternative evaluation approaches when annotations are unreliable.
3. The experimental design is clearly explained with detailed implementation descriptions for each method. The results are presented systematically with consistent metrics and visualization styles, making it easy to compare different methods' strengths and limitations. The inclusion of both high-level performance summaries and detailed analysis helps readers understand the practical implications of the findings.

**Weaknesses:**

1. The evaluation relies heavily on a small set of commonly used datasets, with only 2 datasets for cross-platform analysis and 3 for cross-omics evaluation. The paper fails to examine crucial scenarios such as large-scale retrieval at the million-cell level, rare cell type detection, and performance under noisy or missing data conditions. Essential ablation studies on key components like feature selection strategies and embedding dimensions are missing. Moreover, there is no analysis of computational efficiency, memory usage, or scalability, which are critical factors for practical deployment.
2. The evaluation framework largely adopts standard metrics without introducing domain-specific innovations. The proposed "label-free" metrics are merely basic consistency measurements commonly used in information retrieval, lacking novelty. The paper fails to provide insightful analysis of why traditional methods perform competitively in certain scenarios, and there is no investigation of failure patterns or systematic error analysis. The potential for method combinations or improvements is left unexplored, limiting the paper's technical contributions.
3. The paper does not introduce new challenging datasets or establish novel evaluation protocols that could drive future research. It lacks the necessary infrastructure for community adoption, such as a standardized evaluation pipeline or leaderboard system. Critical components expected in a benchmark paper are missing, including a comprehensive codebase, clear protocol for evaluating new methods, detailed experimental settings in an appendix, and a systematic maintenance plan. Without these elements, the work is unlikely to become a reference benchmark in the field.
4. The study fails to provide clear guidelines for method selection in different scenarios, which would be valuable for practitioners. There is no analysis of the trade-offs between accuracy and computational resources, which is crucial for real-world applications. The paper inadequately investigates robustness to batch effects and data quality variations, common challenges in single-cell analysis. Furthermore, the potential complementary strengths among different methods are not explored, missing an opportunity to guide the development of more effective hybrid approaches.

**Questions:**

Please refer to Weaknesses.

---

### Note · Authors · 2024-11-22

I have read and agree with the venue's withdrawal policy on behalf of myself and my co-authors.